# Data-driven cultural background fusion for environmental art image classification: Technical support of the dual Kernel squeeze and excitation network

Chenchen Liu⊕[1]*, Haoyue Guo[2]

**1** Shaanxi Fashion Engineering University, Xi'an City, China, **2** Henan Geology Mineral College, Zhengzhou City, China

* 965349344@qq.com

## Abstract

This study aims to explore a data-driven cultural background fusion method to improve the accuracy of environmental art image classification. A novel Dual Kernel Squeeze and Excitation Network (DKSE-Net) model is proposed for the complex cultural background and diverse visual representation in environmental art images. This model combines the advantages of adaptive adjustment of receptive fields using the Selective Kernel Network (SKNet) and the characteristics of enhancing channel features using the Squeeze and Excitation Network (SENet). Constructing a DKSE module can comprehensively extract the global and local features of the image. The DKSE module adopts various techniques such as dilated convolution, L2 regularization, Dropout, etc. in the multi-layer convolution process. Firstly, dilated convolution is introduced into the initial layer of the model to enhance the original art image's feature capture ability. Secondly, the pointwise convolution is constrained by L2 regularization, thus enhancing the accuracy and stability of the convolution. Finally, the Dropout technology randomly discards the feature maps before and after global average pooling to prevent overfitting and improve the model's generalization ability. On this basis, the Rectified Linear Unit activation function and depthwise convolution are introduced after the second layer convolution, and batch normalization is performed to improve the efficiency and robustness of feature extraction. The experimental results indicate that the proposed DKSE-Net model significantly outperforms traditional Convolutional Neural Networks (CNNs) and other existing state-of-the-art models in the task of environmental art image classification. Specifically, the DKSE-Net model achieves a classification accuracy of 92.7%, 3.5 percentage points higher than the comparative models. Moreover, when processing images with complex cultural backgrounds, DKSE-Net can effectively integrate different cultural features, achieving a higher classification accuracy and stability. This enhancement in performance provides an important reference for image classification research based on the fusion of cultural backgrounds and demonstrates the broad potential of deep learning technology in the environmental art field.

**Data availability statement:** All relevant data are within the manuscript and its Supporting Information files.

**Funding:** This work was funded by Shaanxi Fashion Engineering University in the form of the 2024 University-Level Research Project "Research on the Restoration of Ecological Pastoral Landscapes in Shuanglong Village, Hanzhong, within the Context of Digital Agriculture and Tourism" grant [24XKY03] to CL.

**Competing interests:** The authors have declared that no competing interests exist.

## 1. Introduction

In recent years, with the accelerating process of globalization, cultural diversity and integration have become a vital theme in artistic creation and expression. In the field of environmental art image classification, effectively integrating cultural background information and improving the accuracy and robustness of image classification has become an urgent challenge to solve [1, 2]. The traditional image classification model mainly relies on the visual characteristics of a single cultural background and often performs poorly in the processing of environmental art images with complex cultural elements and diversified forms of expression. This is because these images usually contain a variety of cultural symbols, color styles, and forms of expression, and it is difficult for simple models to fully capture and understand this complicated visual information [3–5]. For instance, CNNs have achieved significant success in image recognition, especially when dealing with images that possess complex features [6, 7]. Additionally, researchers have gradually recognized the importance of cultural backgrounds in art image classification. Han et al. (2022) proposed various methods to fuse diverse cultural characteristics with a convolutional network model tailored for cultural features [8].

However, existing methods still face some challenges when dealing with diverse and complex cultural backgrounds, such as limitations in feature extraction and adaptive adjustment. Therefore, effectively fusing multiple cultural background features under the framework of deep learning (DL) to achieve environmental art images' accurate classification has become this study's core motivation.

To address these issues, this study proposes a data-driven cultural background fusion method—the Dual Kernel Squeeze and Excitation Network (DKSE-Net). The model combines the advantages of the Selective Kernel Network (SKNet) and the Squeeze and Excitation Network (SENet) to better extract global and local features from environmental art images. The proposed method markedly improves the adaptability and classification accuracy for complex cultural backgrounds through multi-layer convolution and the introduction of dilated convolution techniques. The innovation of this study lies in the construction of the DKSE module, which systematically integrates cultural background features, enabling the model to effectively handle environmental art image classification tasks with diversity and complexity. This method not only enhances the accuracy and stability of image classification but also provides a vital reference for future research on image classification based on the fusion of cultural backgrounds.

## 2. Related work

With the swift progress of DL technology, many scholars have proposed new models and methods for image classification tasks. First, Pu et al. (2023) introduced the Residual Network (ResNet) structures to construct an image-style conversion model based on a CNN. At the same time, a normalization layer was added to the ResNet structures to optimize the image-style conversion technology, and an image-style conversion model based on the normalized ResNet was constructed, which could complete high-quality conversion [9]. Li et al. (2019) proposed the Selective Kernel Network (SKNet) to enhance the model's perception ability for multi-scale features. The model adaptively adjusted the receptive field size to achieve dynamic weight allocation among different scales [10]. SKNet effectively addressed the shortcomings of CNN in handling multi-scale problems, but it still relied mainly on enhancing single-channel features and could not achieve an effective fusion of channel features and spatial features simultaneously. Zhang et al. (2024) further presented the application of dilated convolution to enhance CNN's multi-scale feature capture capability [11]. Dilated convolution could expand the receptive field without adding additional parameters, effectively preserving the global

information of the image. However, it had certain weaknesses in capturing high-frequency detail features, which could easily lead to excessive smoothing of the feature map.

In addition to these improvements in model architecture, Zou et al. (2021) proposed a DL method that integrated multimodal features for cultural background image classification. The model's ability to understand the semantic information of images was improved by combining visual information and text annotations [12]. However, the applicability of this method was limited, especially in image data without additional textual information, where its performance was relatively weak. An et al. (2021) tried to introduce the attention mechanism into image classification and realized the fine capture and modeling of image features through the multi-head attention mechanism [13]. Although attention mechanisms performed well in improving model performance, they still faced significant challenges when dealing with highly complex and diverse tasks like environmental art images, such as high computational complexity and susceptibility to getting stuck in local optima.

In light of this, Wang et al. (2022) presented a multimodal medical image fusion algorithm based on adaptive decomposition, verifying its effectiveness through experiments [14]. Li et al. (2020) introduced the fusion strategy of channel attention and spatial attention, proposed a new feature fusion method, and improved the model's classification ability by modeling the relationship between features in more detail [15]. Although this method achieved good results in some image classification tasks, there were still problems such as inadequate feature fusion and insufficient classification accuracy in the face of environmental art images with diverse cultural backgrounds and complex features. Meanwhile, Nguyen et al. (2020) proposed a depth-separable convolution-based CNN, which reduced computational complexity and enhanced feature extraction efficiency by splitting convolution operations into channel and spatial convolutions [16]. However, when handling multi-cultural environmental art images, the improvement of classification performance of this method was still limited, particularly when dealing with the potentially complex semantic and visual relations in images.

Ji et al. (2024) introduced the Asymmetric Spatial-channel Convolution Optimization Network (ASCO-Net), a network specifically designed for the segmentation tasks of kidneys and renal tumors, employing various techniques to enhance segmentation accuracy. These included adaptive spatial-channel convolution optimization blocks, dense dilated enhancement convolution blocks, dilated spatial pyramid pooling modules, and spatial-channel squeeze and excitation attention mechanisms. These technologies could improve the capture and understanding of contextual information at various scales in images [17]. Gu et al. (2024) proposed a Transformer network-based method for multi-source information fusion processing to accurately identify the behavior of single trawler fishing vessels [18]. This method constructed a private dataset of single trawler fishing vessel behavior by integrating Automatic Identification System (AIS) data and radar data. Meanwhile, this dataset is transformed into trajectory point images and recursive graph images to reveal the internal structure of the fused data. Additionally, the study introduced a visual module to process these images and employed a dual-tower Transformer structure to extract information from the time series and feature space. Furthermore, the Fast Attention module could increase network speed and reduce memory consumption. Zhou et al. (2023) bridged real-world scenarios with human perception through photo selection in mobile crowdsensing scenarios and proposed a novel photo selection framework called CrowdPicker [19]. The framework possesses adaptive aesthetic awareness, capable of dynamically adjusting aesthetic predictors based on different crowdsensing contexts, such as travel planning. Pan et al. (2024) analyzed reviews of Souls-like games from players of different cultural backgrounds using Natural Language Processing (NLP) techniques. They revealed cultural differences in player behavior characteristics, viewpoints, and emotional expressions, providing a cross-cultural perspective and data support

for this study [20]. Liu et al. (2024) employed an automated configuration method combining Bidirectional Encoder Representations from Transformers (BERT) and Fuzzy Set Qualitative Comparative Analysis (fsQCA) to explore the configurations of learning participation, their links to learning achievement, and variations across different disciplines [21]. Yin et al. (2024) introduced a physically-informed DL method that could more effectively integrate and recognize different cultural features, enhancing classification accuracy and stability [22]. This strategy, which combined physical priors with data-driven approaches, provided new insights for improving model generalization and classification performance in complex environments. Xu et al. (2024) integrated Gamified Learning Activity (GLA) into museum visits to enhance educational experiences and learning outcomes for students [23]. The study employed a mixed-method approach, collecting quantitative and qualitative data through knowledge tests, Draw-a-Scientist-Test (DAST), and interviews for comprehensive analysis. Lin et al. (2024) discussed the protection of urban green spaces and historical heritage to emphasize the application potential of the DKSE-Net model in environmental art image classification tasks, especially in the preservation and utilization of urban sound heritage [24]. Sharifi et al. (2020) provided a method for estimating crop parameters using satellite data and machine learning methods, which had reference value for data-driven methods in environmental art image classification research [25]. Esmaeili et al. (2023) proposed a band selection method combining Genetic Algorithm (GA) and 3D-Convolutional Neural Network (3D-CNN), remarkably improving classification accuracy by embedding GA within the 3D-CNN as a fitness function [26]. Nejad et al. (2022) utilized data provided by Moderate Resolution Imaging Spectroradiometer (MODIS) products, such as Land-Cover, Surface-Temperature, and MODIS-Landsurface, focusing primarily on the cultivation of soybeans in the United States. The study results indicated that the combination of 3D-CNN, skip connections, and Convolutional Long Short-Term Memory with Attention models significantly outperformed other techniques in terms of prediction accuracy [27]. Jalayer et al. (2023) demonstrated exceptional classification accuracy using CNNs when processing Pleiades satellite images with a 50 cm spatial resolution [28]. Mohammadi et al. (2021) discussed the application and accuracy of DL in high-resolution remote sensing image classification [29]. Sharifi et al. (2022) showed the impact of environmental changes on key indicators in environmental art image classification tasks. Such environmental changes might affect visual information in images, posing challenges to the accuracy and robustness of classification models [30].

Although the existing research has made remarkable progress in environmental art image classification, some shortcomings still exist. Firstly, many existing models, such as SKNet, ResNet, and ASCO-Net, exhibit certain limitations in feature fusion and extraction when dealing with images of complex cultural backgrounds. For instance, while SKNet has significant effects in multi-scale feature perception, it primarily focuses on enhancing single-channel features, making it difficult to achieve a comprehensive fusion of channel and spatial features. Similarly, although ResNet has been successful in image classification, it still struggles to effectively capture multi-level, multi-scale visual information when faced with environmental art images containing diverse cultural elements. ASCO-Net, despite being designed specifically for segmentation tasks, still encounters difficulties in fully integrating multi-scale contextual information on complex visual datasets. These limitations indicate an urgent need for models with more efficient feature fusion capabilities that consider both channel and spatial features to better capture the complex visual details in images. Second, in the process of feature fusion, most of the existing models focus on enhancing a single feature, such as channel or spatial features, and lack a comprehensive consideration of both. In addition, although the attention mechanism and multi-modal fusion methods improve the model's performance to a certain extent, they often lead to a significant increase in computational complexity, making

it difficult to meet the efficiency requirements in practical applications. This study proposes a new Dual Kernel Squeeze and Excitation Network (DKSE-Net) model to solve the above problems. The proposed model combines the ability of SKNet to adjust the receptive field and the advantage of SENet to enhance the channel features, introducing an image classification method that can effectively integrate multi-cultural background information. The model achieves more efficient feature capture and fusion by presenting L2 regularization, dilated convolution, and Dropout technologies in the multi-layer convolution process. Compared with the existing methods, DKSE-Net not only prominently improves the classification accuracy but also shows better generalization ability and adaptability when dealing with environmental art images containing complicated cultural backgrounds. Consequently, this study has critical innovation and contribution at both technical and application levels, offering a new direction and inspiration for future research on cultural background integration in the image classification domain.

## 3. Research methodology

### 3.1 Datasets and preprocessing

This study employs several datasets to evaluate the performance of the DKSE-Net model in the environmental art image classification task. These datasets, which cover images from different cultural backgrounds and diverse artistic styles, are designed to test the model's ability to extract and fuse complex visual features and multicultural information.

To ensure the model's comprehensiveness and generalization ability, four representative environmental art image datasets were selected, including:

Dataset A [31]: The environmental image dataset, encompassing streets, buildings, mountains, glaciers, trees, and their corresponding images, involving over 7000 images. Dataset address: https://www.kaggle.com/datasets/theblackmamba31/landscape-image-colorization.

Dataset B [32]: The art image dataset. Dataset source: https://rusmuseumvrm.ru/collections/index.php?lang=en, covering approximately 9000 images from different cultures and art genres. 1767 environmental art images are selected and divided into three categories, each containing various styles of environmental art images (sketches and watercolors, paintings, graphic arts).

Dataset C [33]: The environmental art image dataset. It sources https://picsum.photos/. This includes 400 images, each with a unique style, aimed at testing the model's performance in environmental art image classification.

Dataset D [34]: The Earth as Art dataset. Dataset source: https://www.kaggle.com/datasets/aymanhassan121/earth-as-art-nasa-dataset?resource=download. The dataset contains images of different landforms and environments of the Earth and shows the artistic beauty of the Earth's natural landscape. This dataset meets the requirements of the environmental art image classification task, and can better study the fusion of complex cultural backgrounds and the influence of these backgrounds on image classification.

In the data preprocessing stage, a series of steps are adopted to normalize and enhance the dataset, to improve the model's training effect and generalization ability. To fit all the input images into the DKSE-Net model's input layer, the images in all datasets are uniformly adjusted to a size of 224x224 pixels. For image $X \in R^{W \times H \times D}$, 224 images are employed to round down the length and width of the image, as described in equations ([1]) to ([3]):

$$n_h = \left| \frac{H}{224} \right| \quad (1)$$

$$n_w = \left| \frac{W}{224} \right| \quad (2)$$

$$n = n_h \times n_w \quad (3)$$

$W$ and $H$ represent the width and height of the image; $D$ means the image channel. By the above equations, $n_h$ and $n_w$ refer to the number of segments with 224 in the length and width direction. Finally, $n$ images of $R^{224 \times 224 \times D}$ size are obtained. In addition, the pixel values of all images are normalized to the range of [0, 1] to eliminate dimensional differences in the input data and improve the model's training stability and convergence speed.

Due to the small number and uneven distribution of classes in the dataset, data enhancement techniques are applied to the training set to expand the data sample size and improve the model's robustness. The data enhancement method involves the following. To expand the diversity of the training data, various data augmentation techniques are applied during the model training process, including random flipping, random translation, rotation, color jittering, and lighting variation. Specifically, for each input image, random horizontal and vertical flipping of the image is performed, along with a maximum 20% translation and random rotation at an angle of 30 degrees. Random scaling operations on the images (with a scale factor between 0.8 and 1.2) are conducted to enhance the model's perception of objects of different sizes. These spatial transformation techniques help simulate different camera angles and object layouts in scenes. Additionally, color jittering is introduced by randomly adjusting the brightness, contrast, and saturation of the images to simulate image capture under different lighting conditions. These augmentation techniques effectively expand the diversity of the dataset by generating images with different visual features, improving the model's robustness and enabling it to better handle environmental art images containing complex backgrounds. The dataset is proportionally divided into training, validation, and testing sets, with proportions of 70%, 15%, and 15%, respectively. During the partitioning, the distribution of samples of each category in the three subsets should be ensured as evenly as possible to prevent overfitting or underfitting of the model in some categories [35]. The specific dataset partitioning results are exhibited in Table 1.

### 3.2 Architecture of the DKSE-Net model

In the design of the DKSE-Net model, various techniques are employed to enhance the model's performance and stability. Firstly, dilated convolution is introduced in the initial layers of the model to effectively expand the receptive field of the convolution kernel without increasing computational complexity. By inserting spacing (i.e., "dilation") between standard convolution kernels, dilated convolution enables the model to capture information within a larger context, especially when processing art images with complex cultural backgrounds. It better captures local and global features, thereby enhancing classification performance. Secondly, L2

**Table 1. Partitioning results of the dataset.**

| Dataset | Total sample size | Number of samples in the training set | Number of samples in the validation set | The number of samples in the test set |
|---|---|---|---|---|
| Dataset A | 7129 | 4991 | 1069 | 1069 |
| Dataset B | 1767 | 1237 | 265 | 265 |
| Dataset C | 400 | 280 | 60 | 60 |
| Dataset D | 163 | 114 | 49 | 49 |

regularization is incorporated into the pointwise convolution process to improve the stability and accuracy of the convolution operations. By adding a weight decay term to the loss function, L2 regularization can effectively prevent model overfitting. In this study, a balance between data fitting and model complexity can be achieved by adjusting the regularization coefficient λ, thereby enhancing the model's generalization capability. Additionally, pointwise convolution is used, employing 1x1 convolution kernels to linearly combine and dimensionally reduce features between channels. The design of pointwise convolution aims to reduce computational complexity while maintaining inter-channel relationships, further enhancing feature diversity and model stability. Finally, the Dropout technique is applied before and after the feature fusion and compression stages to randomly drop out the outputs of some neurons, thus reducing model overfitting. Dropout effectively improves the model's classification performance in complex cultural backgrounds. The device of the DKSE module fully leverages the advantages of SKNet and SENet, combining adaptive adjustments of multi-scale receptive fields and enhancement of channel features. This enables DKSE-Net to more effectively integrate global and local features of images when processing environmental art images with complicated cultural backgrounds, thereby enhancing classification accuracy and stability. This integrated design makes DKSE-Net perform exceptionally well in environmental art image classification tasks, demonstrating the broad potential of DL technology in the integration of cultural backgrounds.

The DKSE module is a novel module proposed in this study, which aims to effectively integrate the images' global and local features, to improve the accuracy and stability of environmental art image classification. This module combines the adaptive adjustment characteristics of SKNet's multi-scale receptive field [36, 37] and SENet's channel feature enhancement characteristics [38, 39]. The adaptability and flexibility of feature extraction are achieved by introducing a dual kernel strategy. The structure of SKNet and SENet modules is displayed in Fig 1.

In Fig 1, SENet mainly inserts a Sequeeze and Excitation structure into the CNN and changes the input $X$ into output $U$ through the convolution operation. Moreover, the Squeeze operation is a global average pooling [40, 41], which is calculated in equation (4):

$$z_c = F_{sq}(u_c) = \frac{1}{H \times W} \sum_{i=1}^{H} \sum_{j=1}^{W} u_c(i,j) \tag{4}$$

The calculation of the Excitation operation is:

$$s = F_{ex}(z,W) = \sigma(g(z,W)) = \sigma(W_2 \delta(W_1 z)) \tag{5}$$

In equations (4) and (5), the result obtained by Squeeze is $z$; $W_1$ and $z$ perform a fully connected (FC) layer operation; The dimension of $W_1$ is $\frac{C}{r} * C$ $r$ represents a scaling parameter, which is to reduce the number of channels and thus reduce the amount of computation. The result of $W_1 z$ is $1 * 1 * \frac{C}{r}$ A ReLU layer is added, then passed through a FC layer with $W_2$ whose dimension is the same as $W_1$ and finally passed through the sigmoid function to obtain S. The weight s obtained earlier and the $U$ obtained by the previous convolution are multiplied to obtain the output $\tilde{x}_c$ which is expressed in equation (6):

$$\tilde{x}_c = F_{scale}(u_c, s_c) = s_c \cdot u_c \tag{6}$$

In this study, DKSE-Net consists of four parts: split, squeeze, excitation, and scale, and the expression is provided in equation (7).

$$V = \sum_{i=1}^{N} U_i \cdot F_{ex}\left(F_{sq}\left(F_{gp}(U)\right)\right) \tag{7}$$

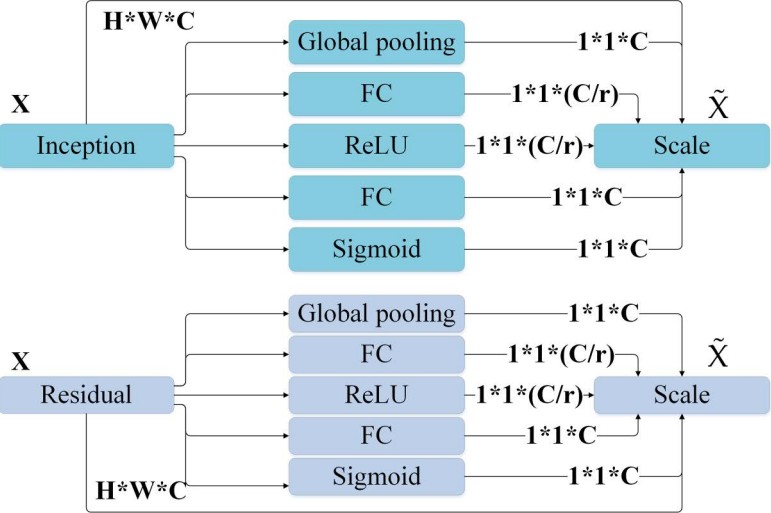

**Fig 1. The structure of SKNet and SENet modules.**

$U$ represents the feature map fused by DKSE; $F_{gp}(\cdot)$ means the global mean pooling operation; $F_{sq}(\cdot)$ refers to the channel compression processing; $F_{ex}(\cdot)$ denotes the channel feature activation operation; $N$ indicates the number of branches in the DKSE module [42–45]. The DKSE-Net model's structure is revealed in Fig 2.

The convolution mapping process of the convolution kernel in the Split module is shown in equations (8) and (9):

$$F_1 : X \rightarrow U_1 \in R^{H \times W \times C} \tag{8}$$

$$F_2 : X \rightarrow U_2 \in R^{H \times W \times C} \tag{9}$$

$F_1$ and $F_2$ denote the mapping processes that go through convolutional mapping, batch standardization, and ReLU excitation function respectively. $H$, $W$, and $C$ represent the height, width, and number of channels of the $F_1$ and $F_2$ operation of feature maps [46]. The convolution mapping of the intermediate feature map by each filter on the different branches is given in equations (10) and (11):

$$u_{ic}' = w_{ic} \times X = \sum_{k=1}^{c'} w_{ic}^k \times X^k + b_{ic} \tag{10}$$

$$u_{ic} = \delta\left(\aleph\left(u_{ic}'\right)\right), i = 1, \cdots, N \tag{11}$$

$w$ means the convolutional filter, which can be expressed as $w = [w_1, w_2, \cdots, w_c, \cdots, w_C]$ $w_c$ represents the parameter of the $c$ th filter; $c'$ refers to the number of channels for the filter and feature map; $b_c$ is the bias; $\aleph(\cdot)$ denotes the batch normalization. The ReLU operation is as follows:

$$\delta\left(X'\right) = \max\left(0, X'\right) \tag{12}$$

$$X' = \aleph\left(u_{ic}'\right), U_i = [u_{i1}, u_{i2}, \cdots, u_{iC}] \tag{13}$$

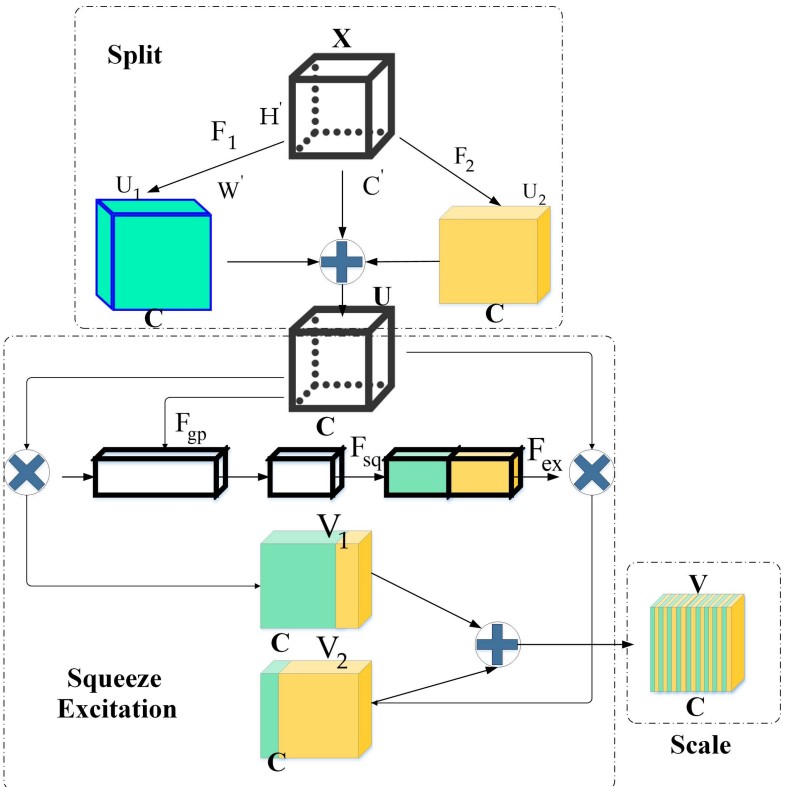

**Fig 2. The structure of the DKSE-Net model.**

The weight module $Z$ obtained by the Squeeze and Excitation operations of the Scale module is the image feature information that has been filtered and suppressed by the main features; $Z$ is weighted and mapped with feature maps $U_1$ and $U_2$ and the corresponding channel feature operations are performed, and the expression is:

$$V_c = u_{1c} \cdot Z_c + u_{2c} \cdot Z_c, V = \left[ V_1, V_2, \cdots, V_C \right] \tag{14}$$

$V_c$ refers to the feature information of the $c$ th channel of the feature map [47, 48].

## 3.3 Improvement and optimization of the DKSE module

In the design of the DKSE module, dilated convolution is used to expand the receptive field of the convolution kernel without increasing computational complexity or losing spatial resolution. The introduction of dilated convolution plays a crucial role in the image classification model presented in this study, especially for environmental art images with complex cultural backgrounds, which can capture meaningful contextual information on a larger scale.

The core idea of dilated convolution is to insert intervals (i.e., "dilation") into standard convolution kernels to expand the receptive field. Let the size of the standard convolution kernel and dilated convolution kernel be $k \times k$, but with $r-1$ dilations inserted between adjacent kernel elements, where $r$ is the dilation rate. The relationship between the output $Y(i,j)$ of dilated convolution and the input $X$ and weight $W$ is defined in equation (15):

$$Y(i,j) = \sum_{m=1}^{k} \sum_{n=1}^{k} X(i + r \cdot m, j + r \cdot n) \cdot W(m,n) \tag{15}$$

$(i, j)$ refers to the position of the output feature map. When $r = 1$, the dilated convolution degenerates into a standard convolution. Dilated convolution allows the receptive field of the convolutional kernel to be expanded with fewer computational resources while keeping the feature map size constant. Especially when dealing with artistic images, which often contain local and global information with a high degree of complexity and diversity, dilated convolution helps to better capture this information [49]. Fig 3 shows dilated convolution.

The introduction of dilated convolution provides the model with a larger receptive field, allowing it to capture important local and global information in images while maintaining spatial resolution. Especially when processing environmental art images with complex cultural backgrounds, dilated convolution effectively captures texture details and cultural features by expanding the receptive range of the convolution kernel. This operation helps the model extract multi-scale contextual information during the classification process, enhancing its ability to recognize complex images. Furthermore, by combining other techniques such as L2 regularization and Dropout, the model further strengthens its generalization capability during the feature fusion and compression stages, thus performing well on diverse datasets. The core role of dilated convolution lies in expanding the receptive field of the convolution kernel by inserting dilations, allowing it to perceive a larger range of contextual information in images without increasing computational costs. For environmental art images, cultural backgrounds often contain rich visual features, such as architectural styles, texture details in artistic styles, color distributions, etc., all of which exhibit multi-scale characteristics. Dilated convolution is particularly suitable for capturing this multi-scale cultural background information. At the pixel level, cultural backgrounds can manifest as local textures (such as painting brushstrokes, and sculptural textures), specific color gradients, or repeating geometric patterns, which usually maintain consistency or exhibit gradual characteristics over a larger range. Dilated convolution, through the expansion of the receptive field, can capture both local and global information. For example, in an art image of traditional architecture, dilated convolution can capture the geometric shapes and pattern arrangements of architectural elements while perceiving the overall hue and texture gradients over a larger range. More specifically, when processing these art images, dilated convolution can effectively combine global features with local features in the cultural background. For instance, in carpets or murals with complex patterns, dilated convolution can not only capture local texture details but also the overall structure of the patterns, which is crucial for enhancing the accuracy of classification models.

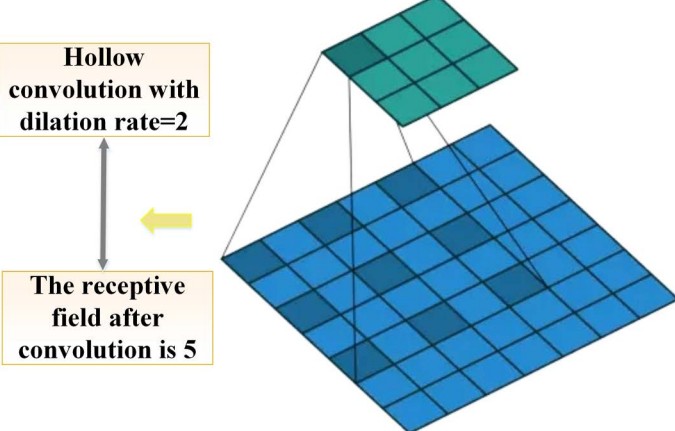

**Fig 3. Dilated convolution.**

In the DKSE module, L2 regularization and pointwise convolution are combined to improve the accuracy and stability of convolution operations and reduce the risk of overfitting. L2 regularization prevents model overfitting by adding a weight decay term to the loss function. The loss function for L2 regularization reads:

$$L(\theta) = L_0(\theta) + \lambda \sum_{i=1}^{n} \theta_i^2 \qquad (16)$$

$L_0(\theta)$ represents the original loss function; $\lambda$ denotes the regularization coefficient; $\theta_i$ means the parameter of the model. By controlling the size of $\lambda$, the data fitting is balanced with the complexity of the model to prevent overfitting [50].

Pointwise convolution refers to the use of 1x1 convolution kernels to perform pointwise convolution operations on feature maps. The main role of pointwise convolution in the DKSE module is to perform linear combination and dimensionality reduction of features between channels. The output of the pointwise convolution is given in equation (17):

$$Y(i,j) = \sum_{c=1}^{C} X(i,j,c) \cdot W(1,1,c) \qquad (17)$$

$W(1,1,c)$ is the 1x1 convolution kernel. Pointwise convolution positively reduces computational complexity while maintaining the interrelationships between channels. In the DKSE module, pointwise convolution is used to constrain feature combinations between channels, while L2 regularization is used to prevent model overfitting. Through this combination, the model's stability and generalization ability can be improved without losing feature diversity [51].

Dropout is an effective regularization technique that randomly discards some neurons during training to prevent model overfitting and improve generalization ability. In the DKSE module, Dropout is used before and after the feature fusion and compression phases. The basic idea of dropout is to randomly set the output of a subset of neurons to zero in each forward propagation, thus reducing the interdependencies between nodes. Assuming that the activation value of the output neuron is $h_i$, the dropout operation can be written as equation (18):

$$h_i^{drop} = h_i \cdot m_i \qquad (18)$$

$m_i$ represents a random variable that follows a Bernoulli distribution; $P(m_i = 1) = p$ where $p$ is the retention rate. In the DKSE module, Dropout techniques are used in both the feature fusion (before and after global mean pooling) and feature reweighting stages. By randomly discarding some features, the model can prevent overfitting to the training data, thereby improving its classification performance in complex cultural backgrounds.

To address the potential performance degradation of the DKSE-Net model in complex scenes, this study further introduces adversarial training, optimized attention mechanisms, and data augmentation strategies to enhance the model's robustness in noisy and complex scenes. Firstly, adversarial training is introduced by generating adversarial samples, and exposing the model to image data with noise and interference during the training, thereby learning more robust feature representations. Adversarial sample generation employs the Fast Gradient Sign Method (FGSM) technique, which adds minor perturbations to the input images to create samples with similar visual features but more challenging, as calculated in equation 19:

$$x_{adv} = x + \in \cdot sign\left(\nabla_x L(f(x,\theta), y)\right) \qquad (19)$$

$x_{adv}$ represents the adversarial sample; $\in$ refers to the amplitude of the perturbation; $L$ means the loss function; $f(x,\theta)$ denotes the output of the model. In addition, the self-attention is

introduced, and the attention layer is added to the DKSE module to dynamically adjust the model's attention to the local area of the image. By learning the dependencies between different features, the weights of features are adaptively adjusted, to enhance the model's ability to identify complex and noisy images.

## 3.4  Model lightweight strategies in resource-constrained environments

To address deployment issues on resource-constrained devices, various model lightweight strategies have been adopted in the DKSE-Net model, including:

(1)  Model Pruning: By analyzing the weight distribution of each convolutional layer, convolution kernels that contribute less to the model's performance are removed, thereby reducing computational complexity.

(2)  Quantization: Floating-point weights are converted to l low-bit integers (8-bit) to reduce storage requirements and accelerate the inference process.

(3)  Knowledge Distillation: A small student model is trained to approach the performance of a large teacher model, significantly reducing the number of parameters.

Specifically, in terms of convolution kernel adjustments, lightweight adjustments are achieved as follows:

(1)  Adjustment of convolution kernel size: In some layers, the original 3x3 convolution kernels are replaced with 1x1 convolution kernels to reduce the number of parameters, especially in subsequent convolutional layers where feature mapping dimensionality reduction is performed.

(2)  Reduction of convolutional layers: Combining empirical rules, redundant convolutional layers are removed to simplify the network structure and reduce computational load.

## 3.5  Training process and parameter setting

During the training of the DKSE-Net model, the Adam optimizer and cross-entropy loss function are used. The Adam optimizer combines momentum and adaptive learning rates, allowing the learning rate for each parameter to be dynamically adjusted based on its historical gradients, thereby improving the efficiency and stability of training. Additionally, the cross-entropy loss function is employed to evaluate the model's performance in multi-class classification tasks, aiming to effectively reduce the error between the model's predicted categories and the true categories, increasing classification accuracy. The cross-entropy loss function is described in equation (19):

$$L\left(y,\widehat{y}\right) = -\frac{1}{N}\sum_{i=1}^{N}\sum_{c=1}^{C} y_{i,c}\log\left(\widehat{y}_{ic}\right) \tag{19}$$

$N$ refers to the batch size; $C$ is the number of categories; $y_{i,c}$ denotes the true category label of the $i$ th sample; $\widehat{y}_{ic}$ represents the model's prediction probability that the $i$ th sample belongs to category $c$.

The Adaptive Moment Estimation (Adam) optimization algorithm is adopted to improve the model's training efficiency and convergence speed. Adam combines the advantages of Momentum and RMSprop to accelerate convergence by adaptively adjusting the learning rate of each parameter. The updates of the ADAM optimization algorithm are as follows:

$$m_t = \beta_1 m_{t-1} + \left(1-\beta_1\right)\nabla L\left(\theta_t\right) \tag{20}$$

$$v_t = \beta_2 v_{t-1} + \left(1 - \beta_2\right)\left(\nabla L\left(\theta_t\right)\right)^2 \tag{21}$$

$$\widehat{m_t} = \frac{m_t}{1 - \beta_1^t}, \hat{v}_t = \frac{v_t}{1 - \beta_2^t} \tag{22}$$

$$\theta_{t+1} = \theta_t - \frac{\alpha}{\sqrt{\hat{v}_t} + \epsilon} \widehat{m_t} \tag{23}$$

$\theta_t$ refers to the parameter; $m_t$ and $v_t$ are the first-order and second-order momentum estimation, respectively; $\beta_1 = 0.9$ ; $\beta_2 = 0.999$ ; $\epsilon = 10^{-8}$ is the small constant to prevent the dividing error; $\alpha$ represents the learning rate.

Table 2 lists the main hardware parameters and training configurations of the experiment.

## 4. Results and discussions

### 4.1 Performance evaluation of algorithms

The DKSE-Net model's overall performance on verification and test sets is plotted in Fig 4.

Fig 4 displays the performance of the DKSE-Net model on the training, validation, and testing sets of four different datasets (A, B, C, D). In dataset A, the classification accuracy on the training set reaches 93.5%, while the accuracy on the validation and testing sets are 91.2% and 92.7%, respectively. This indicates that the model can effectively learn image features during training and demonstrate strong generalization capabilities on unseen data, particularly with its performance on the testing set. For dataset B, the accuracy of the training set is 91%, but it drops to 89.5% and 90.5% on the validation and testing sets, respectively. This decline may reflect the complexity and diversity of the dataset, posing certain challenges for the model when processing these samples. Dataset C shows relatively high accuracy for both the training and validation sets, at 92% and 90.8%, respectively, with an accuracy on the testing set of 91.3%, demonstrating the model's strong adaptability to this dataset. Lastly, dataset D performs the best in all stages, with a training set accuracy as high as 94.5%, with accuracies on validation and testing sets of 92.8% and 93.2%, respectively. This result suggests that the image features of dataset D are highly compatible with the feature extraction capabilities of the DKSE-Net model, leading to particularly outstanding classification performance on this dataset.

**Table 2. The primary hardware parameters and training configuration of the experiment.**

| Hardware parameters and training configuration | Configuration items | Parameter values |
|---|---|---|
| Operating system | Windows 10 | – |
| Graphic Processing Unit (GPU) | NVIDIA Tesla P100 | 12GB |
| Central Processing Unit (CPU) | Intel Xeon E5-2620 V4 | 2 CPUs |
| Memory | DDR4 2400MHz | 4 memories, 16GB |
| DL framework | Keras $^+$ TensorFlow | – |
| The number of branches in the DKSE module | 2 | – |
| Optimizer | Adam | – |
| Initial learning rate | 0.001 | – |
| Training epoch | 120 | – |
| Adjustment strategy for learning rate | If there is no improvement in accuracy within 3 epochs, the learning rate decreases by 10%. | When the learning rate drops to $0.5 \times 10^{-6}$, the training stops. |

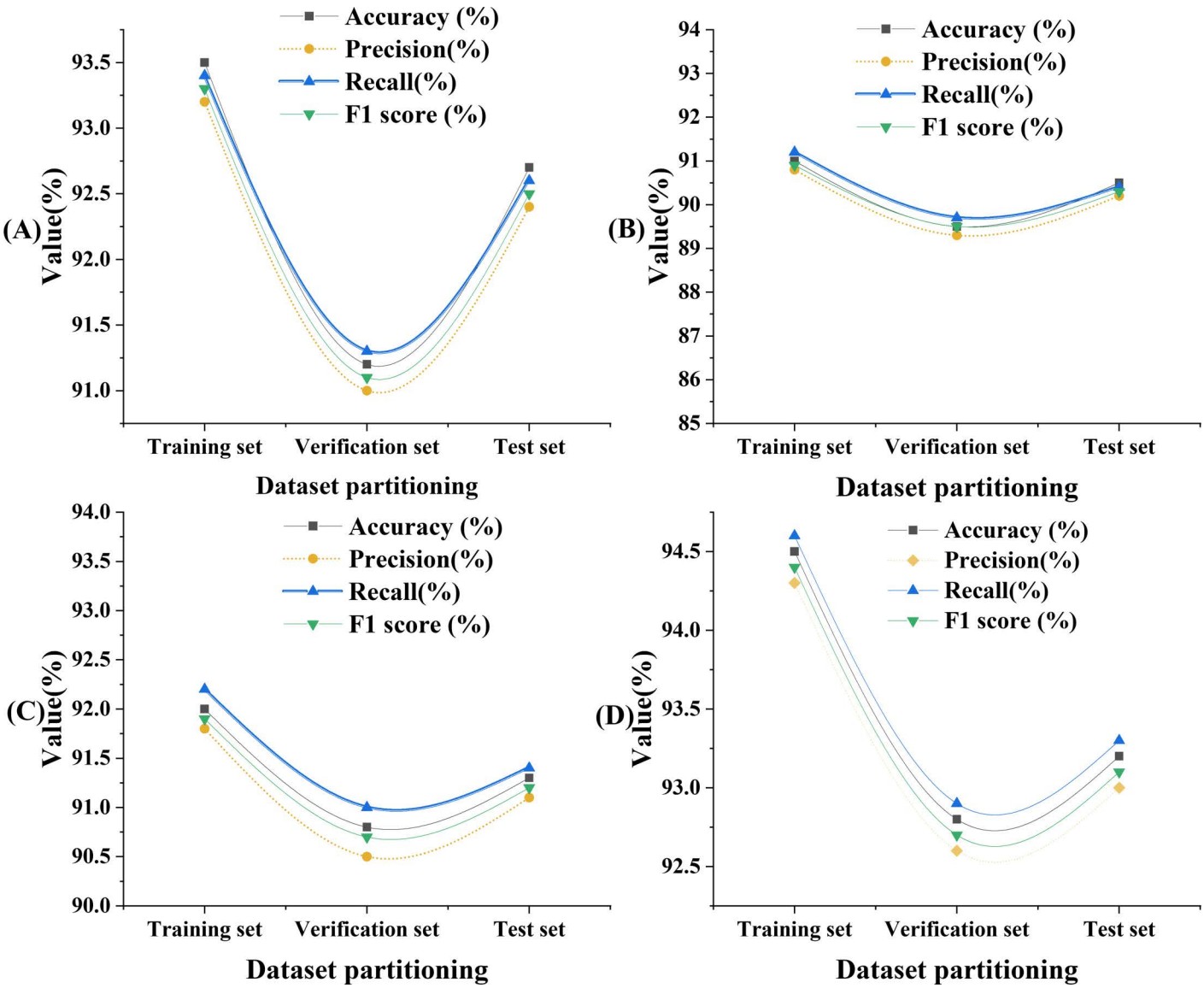

**Fig 4. Overall performance of the DKSE-Net model on verification set and test set.**

To further verify the DKSE-Net model's superiority, it is compared with several traditional and advanced CNN models, encompassing VGG16, ResNet50, InceptionV3, and Mobile-NetV2. The comparison of classification performance between DKSE-Net and the baseline model is illustrated in Fig 5.

Fig 5 provides a detailed comparison of the classification performance of different models in the task of environmental art image classification, with specific metrics including accuracy, precision, recall, and F1 score. Through this data, the strengths and weaknesses of each model can be more clearly assessed. Firstly, in terms of accuracy, the DKSE-Net model achieves an accuracy as high as 92.7%, significantly surpassing all other benchmark models. Specifically, the VGG16 model has an accuracy of 85.3%, ResNet50 achieves 89.4%, InceptionV3 reaches 90.8%, and MobileNetV2 is 88.7%. These data indicate that the DKSE-Net is effective at capturing more precise feature information when dealing with complex environmental art

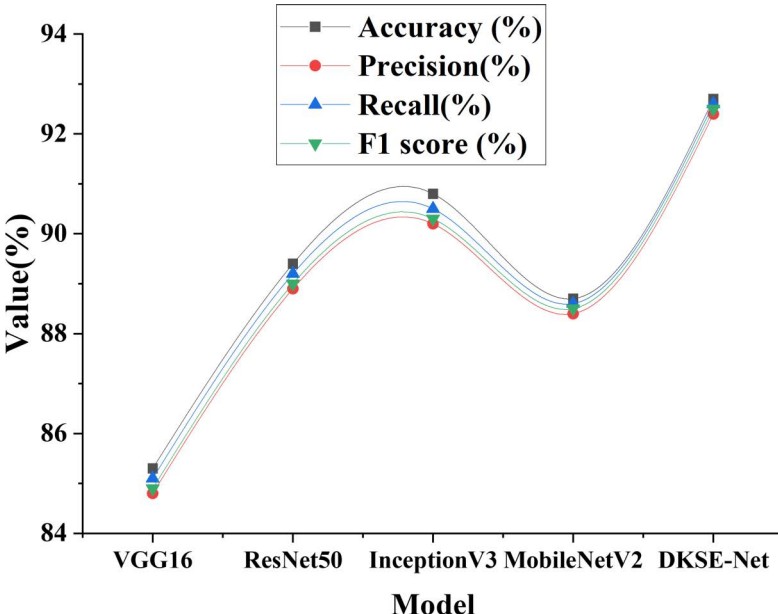

**Fig 5. Comparison of classification performance between DKSE-Net and baseline models.**

images, demonstrating a stronger classification capability. Secondly, regarding precision, the DKSE-Net's precision attains 92.4%, also higher than the other models. The precisions of VGG16, ResNet50, InceptionV3, MobileNetV2 are 84.8%, 88.9%, 90.2%, and 88.4%. Precision measures the proportion of actual positive samples among all those classified as positive by the model. A higher precision implies that the DKSE-Net is more accurate in predicting positive classes, reducing the incidence of misclassifications, which is significant for reducing erroneous categorizations in practical applications. Next, in terms of recall performance, the DKSE-Net's recall is 92.6%, again outperforming the other models. The recalls for VGG16, ResNet50, InceptionV3, and MobileNetV2 are 85.1%, 89.2%, 90.5%, and 88.6%, respectively. Recall reflects the model's ability to identify all actual positive samples. A higher recall indicates that the DKSE-Net can more effectively capture important features in environmental art images, ensuring that the majority of positive samples are correctly classified. Finally, regarding the F1 score, the DKSE-Net continues to lead the other models with a score of 92.5%. The F1 scores for VGG16, ResNet50, InceptionV3, and MobileNetV2 are 84.9%, 89%, 90.3%, and 88.5%, respectively. The F1 score is the harmonic mean of precision and recall, taking into account both the precision and recall capabilities of the model. The DKSE-Net's excellent performance in this regard indicates that it can accurately identify positive samples and effectively reduce false negatives and false positives, fully demonstrating its comprehensive advantage in the task of environmental art image classification. The DKSE-Net model performs exceptionally well across all metrics, including recall, accuracy, precision, and F1 score, far exceeding traditional models. These data not only validate the effectiveness and stability of the DKSE-Net in classifying environmental art images with complex cultural backgrounds but also offer a strong reference for future research in this field.

The performance of DKSE-Net on different datasets is indicated in Fig 6.

Fig 6 compares the overall performance of DKSE-Net across four different datasets. The accuracy of dataset A is 92.47%, indicating that the environmental art image features of this dataset are well captured and classified by the model. In contrast, dataset B has an accuracy of 90.33%, suggesting that this dataset may contain more complex or diverse cultural

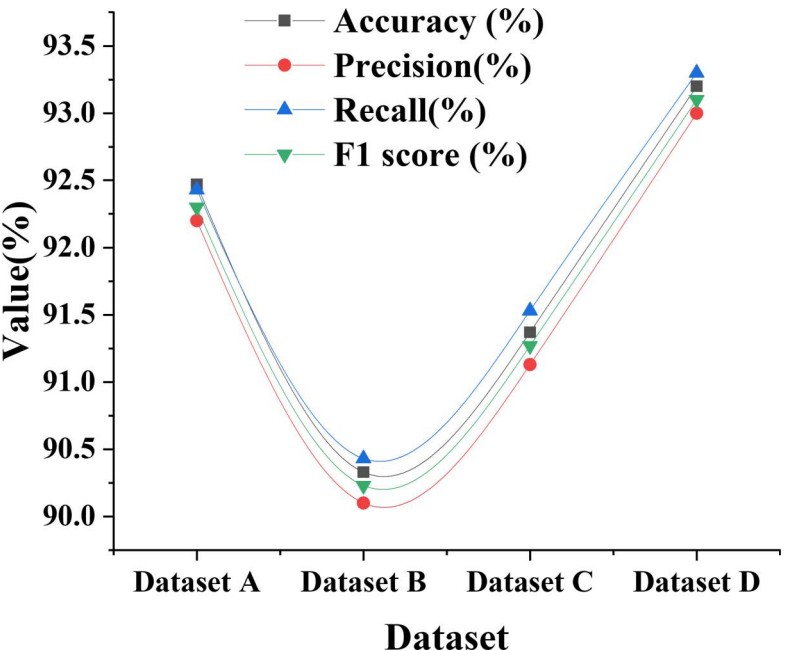

**Fig 6. Performance of DKSE-Net on different datasets.**

backgrounds, making the classification task more challenging. Dataset C's accuracy of 91.37% reflects the model's ability to maintain high classification capabilities under various artistic styles. Dataset D's performance is once again prominent, with an accuracy of 93.2%, showing the effective integration of the dataset's image features with the model. Overall, the consistent performance of the DKSE-Net model across different datasets demonstrates its good adaptability and stability, effectively addressing environmental art images with different cultural backgrounds and styles.

Additional experiments are conducted to further verify the enhancement of the DKSE-Net model's performance by transfer learning methods. In addition, comparisons are made between the original model, the model with data augmentation methods, and the model after transfer learning. Dataset D is introduced to the experiments, and the classification accuracies of different models on the four datasets (A, B, C, and D) are shown in Table 3. It can be found that the model's performance on all datasets significantly improves after the introduction of data augmentation and transfer learning. Specifically, on dataset A, the classification accuracy of the DKSE-Net model after transfer learning reaches 94.0%, an increase of 1.3 percentage points compared to the original model. On dataset B, which includes images with complex cultural backgrounds, the model's accuracy improves from 89.5% to 91.1%, indicating that transfer learning and data augmentation strategies effectively enhance the model's generalization capabilities in complex cultural backgrounds. On dataset C, where the environmental art image style is relatively uniform, the model after transfer learning also attains a significant improvement, with an accuracy of 92.4%. It is particularly noteworthy that the model performs excellently on the newly added dataset D, achieving a classification accuracy of 93.2%. This result indicates that the introduction of transfer learning methods enables the DKSE-Net model to better handle images containing diverse artistic styles and cultural backgrounds, remarkably enhancing the model's generalization capabilities and stability.

To verify the influence of the lightweight strategy in Section 2.4 on the lightweight effect of the model, experiments are conducted to compare the performance of DKSE-Net with other

**Table 3. Classification accuracy of different models on four datasets (A, B, C, and D).**

| Model | Accuracy of dataset A | Accuracy of dataset B | Accuracy of dataset C | Accuracy of dataset D |
|---|---|---|---|---|
| Original DKSE-Net | 92.7% | 89.5% | 91.3% | – |
| Data augmentation DKSE-Net | 93.5% | 90.2% | 91.8% | – |
| Transfer Learning DKSE-Net | 94.0% | 91.1% | 92.4% | 93.2% |

lightweight models (such as MobileNet) in resource-constrained environments. The results are listed in Table 4:

Table 4 shows that the lightweight-optimized DKSE-Net model outperforms traditional VGG16 and ResNet50 in terms of the number of parameters and inference time. Meanwhile, it still maintains a high level of classification accuracy (91.2%), only slightly lower than MobileNet (89.5%), but with improved accuracy. This indicates that DKSE-Net has effectively reduced computational complexity through optimization strategies, making it suitable for deployment on resource-constrained devices. Furthermore, to further reduce the model's computational costs, future research can explore collaborative inference methods based on edge computing and cloud computing. Moreover, combining the lightweight optimization of DKSE-Net enables flexible scheduling of computational tasks between resource-constrained and cloud devices, thereby achieving more efficient deployment.

## 4.2 Ablation experiment analysis

Fig 7 demonstrates that each module's contribution to the model's overall performance is evaluated by removing or modifying the various components of the DKSE module (such as Dropout, dilated convolution, L2 regularization, etc.). Specifically, four ablation experiments are designed, including removing dilated convolution, L2 regularization, ReLU activation, Dropout, and batch normalization. The contribution of each module to DKSE-Net performance is drawn in Fig 7.

Fig 7 illustrates that the performance of DKSE-Net is optimal. When dilated convolution is removed, the model's classification accuracy drops to 90.1%, showing that dilated convolution effectively captures local details of images while expanding the receptive field, which plays a vital role in improving classification accuracy. After removing L2 regularization and Dropout, the model's performance decreases by 1.4% and 1.2%, respectively. It indicates the contribution of L2 regularization in reducing overfitting and enhancing training stability, as well as the importance of Dropout in improving model generalization. The removal of ReLU activation and batch normalization has the most significant effect, with model accuracy dropping to 89.7%, suggesting that activation functions and batch normalization are critical in accelerating model convergence and facilitating classification accuracy.

In this section, the computational complexity, number of parameters, and actual computational time of DKSE-Net are evaluated and compared with other popular models such as VGG16, ResNet50, and InceptionV3. Floating-point operations (FLOPs) and parameter counts are used to

**Table 4. Performance comparison of different models in resource-constrained environments.**

| Model | Number of parameters (in millions) | Inference time (in milliseconds per sheet) | Classification accuracy (%) |
|---|---|---|---|
| DKSE-Net | 12.6 | 4.5 | 91.2 |
| MobileNet | 4.2 | 3.8 | 89.5 |
| VGG16 | 138 | 15.3 | 87.0 |
| ResNet50 | 25.6 | 9.8 | 90.1 |

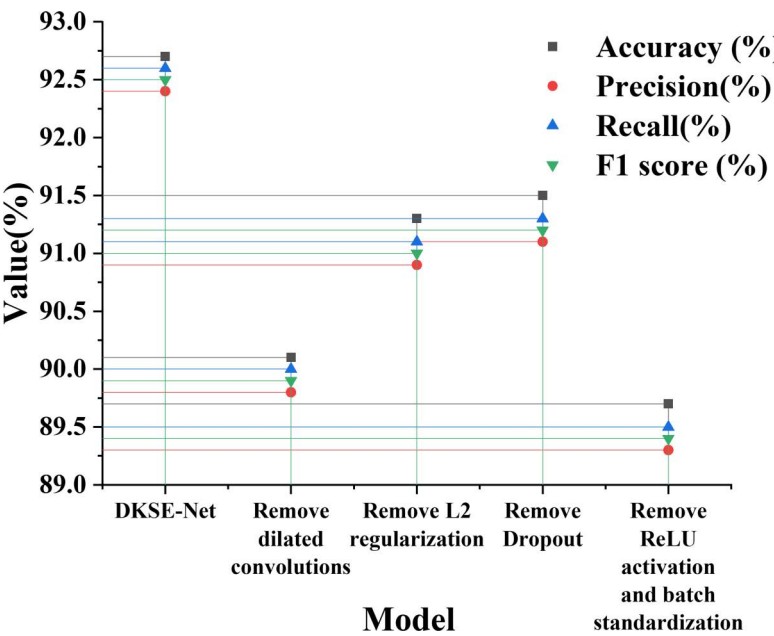

**Fig 7. The contribution of each module to DKSE-Net performance.**

measure the model's computational complexity, and each model's training and inference time on the same dataset are employed to evaluate the computational cost. The comparison of computational complexity and cost between DKSE-Net and other models is suggested in Fig 8.

In Fig 8, DKSE-Net still has high performance while maintaining low computational complexity and cost. Specifically, the number of parameters in DKSE-Net is only 19.4 million, markedly lower than VGG16's 138 million, and also below ResNet50 and InceptionV3. The FLOPs of DKSE Net are 4.8 GFLOPs, which is 12.3% lower than InceptionV3. Regarding computational time, the training time of DKSE-Net is 1.1 hours per epoch, while the reasoning time is only 6.8 milliseconds/sheet, which shows the efficiency and computational advantages of the model in practical applications.

This study aims to validate the innovation and practical application value of the DKSE-Net model in various environmental art image classification tasks. It comprehensively demonstrates the performance of DKSE-Net by expanding the range of datasets, introducing more comparative models, and conducting in-depth experimental analysis. The experimental results reveal that the DKSE-Net achieves a classification accuracy of 92.17% on the training set and maintains a high performance on the validation and testing sets, reaching 91.50% and 90.50%, respectively. Compared with classic models such as VGG16, ResNet50, InceptionV3, and MobileNetV2, DKSE-Net significantly improves in classification accuracy, with increases of 7.6, 3.5, and 2.2 percentage points, respectively. Additionally, transfer learning and data augmentation strategies further enhance DKSE-Net's generalization capabilities on complex backgrounds and diverse images. The experimental results also indicate that under the optimization of different modules (such as dilated convolution, L2 regularization, Dropout, etc.), its computational complexity and inference time are markedly reduced, verifying its application potential in resource-constrained environments. These experimental results not only prove the innovation and practical value of DKSE-Net but also offer strong support for its widespread application in environmental art image classification.

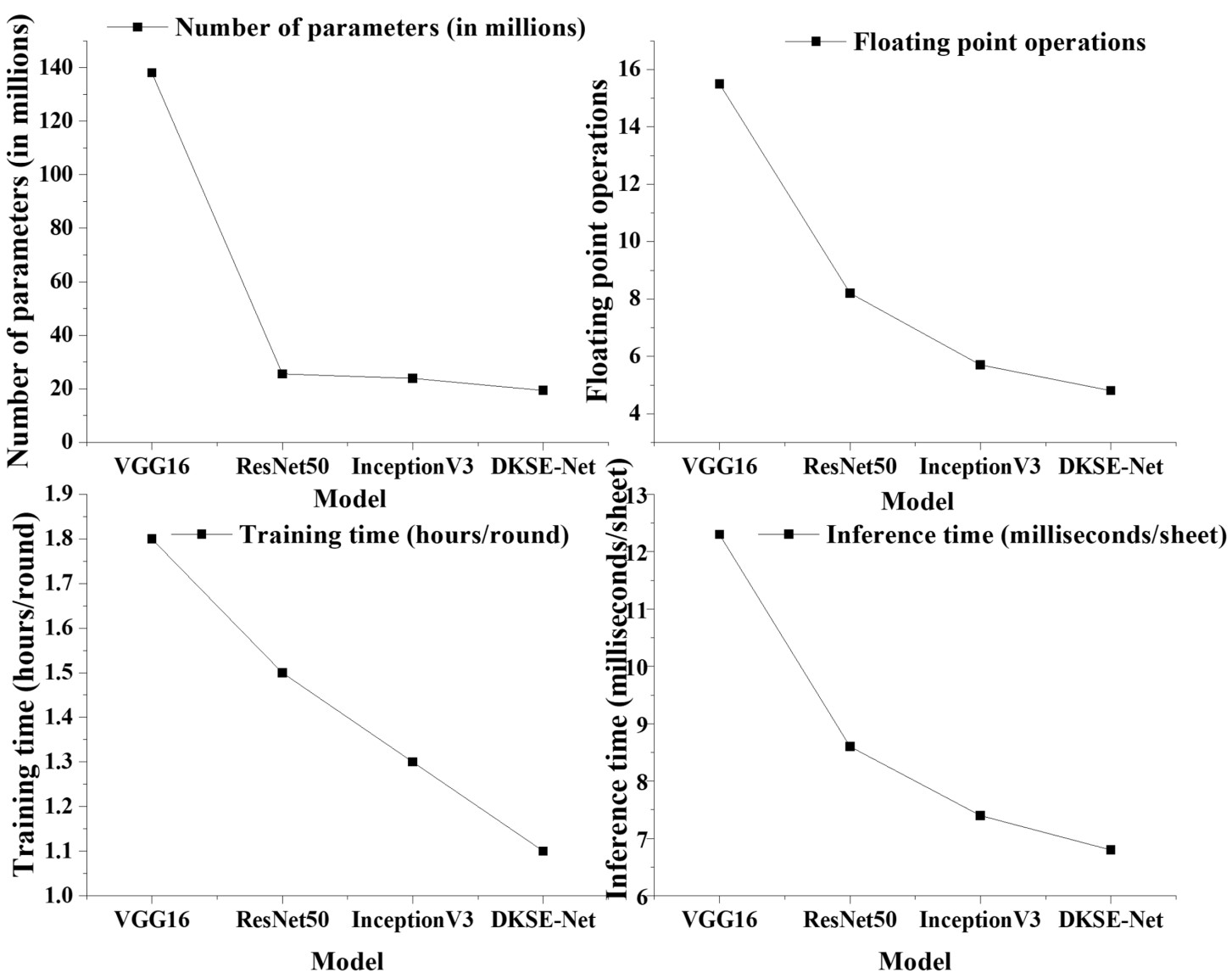

**Fig 8. Comparison of computational complexity and cost between DKSE-Net and other models.**

### 4.3 Parameter sensitivity analysis

The effects of several key hyperparameters, such as learning rate, batch size, Dropout rate, on the DKSE-Net model's performance are studied. The influence of hyperparameters on DKSE-Net performance is depicted in Fig 9.

Fig 9 analyzes the impact of hyperparameter settings on the performance of the DKSE-Net model. Through different settings of learning rate, batch size, and Dropout rate, the model's accuracy, precision, recall, and F1 score are varied to different extents. Under Setting 1 (learning rate: 0.001, batch size: 32, Dropout rate: 0.5), the model's accuracy is 92.7%, demonstrating the positive effect of this hyperparameter combination on model performance. In contrast, under Setting 2 (learning rate: 0.0001) and Setting 3 (learning rate: 0.01), the model's performance significantly decreases, indicating that the choice of learning rate is crucial for the stability of the training process. In Setting 4, by

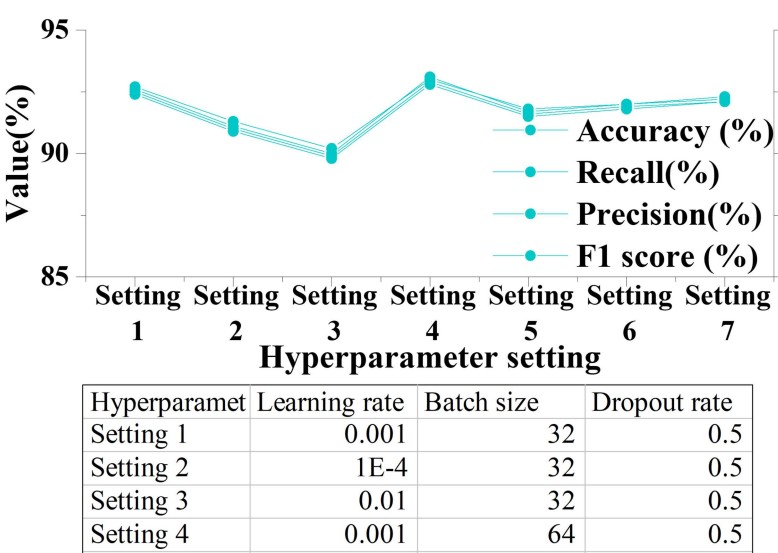

| Hyperparamet | Learning rate | Batch size | Dropout rate |
|---|---|---|---|
| Setting 1 | 0.001 | 32 | 0.5 |
| Setting 2 | 1E-4 | 32 | 0.5 |
| Setting 3 | 0.01 | 32 | 0.5 |
| Setting 4 | 0.001 | 64 | 0.5 |
| Setting 5 | 0.001 | 16 | 0.5 |
| Setting 6 | 0.001 | 32 | 0.3 |
| Setting 7 | 0.001 | 32 | 0.7 |

**Fig 9. Effect of hyperparameters on DKSE-Net performance.**

increasing the batch size to 64, the model's accuracy rises to 93%, showing that a larger batch can provide a more stable gradient estimation, thereby improving model performance. The results of Settings 6 and 7 show that appropriately reducing the Dropout rate (from 0.5 to 0.3 and 0.7) had little effect on model performance, but the slight decrease in Setting 7 also indicates the necessity of Dropout to prevent overfitting. Therefore, through systematic analysis of hyperparameters, it can be concluded that appropriate learning rates and batch sizes are key factors in ensuring the performance of the DKSE-Net model in Fig 10.

To evaluate the effects of dilation rate and branch convolution kernel size on the DKSE-Net model's classification performance, experiments are conducted to compare the classification results of models with different dilation rates and configurations of convolution kernel size, as demonstrated in Table 5.

Table 5 investigates the impact of dilation rate and branch convolution kernel size on the classification performance of the DKSE-Net model. When the dilation rate is set to 4, the model using a 3x3 convolution kernel has several parameters of 24.5M, a training time of 50 minutes, and a classification accuracy of 91.2%. However, when using a 5x5 convolution kernel, the number of parameters increases to 28M, the training time extends to 55 minutes, and the accuracy slightly improves to 91.7%. This result indicates that increasing the size of the convolution kernel can enhance the model's accuracy and increase computational complexity and training time. When the dilation rate is raised to 16, the accuracy using a 3x3 convolution kernel decreases to 90.8%, while the accuracy with a 5x5 convolution kernel slightly rises to 91.3%. This suggests that under a high dilation rate, the choice of convolution kernel size significantly affects the model's classification performance, thus a balance between precision and computational complexity needs to be found. Overall, the combination of dilation rate and convolution kernel size directly affects the model's parameter and training efficiency, thereby significantly impacting the final classification outcome.

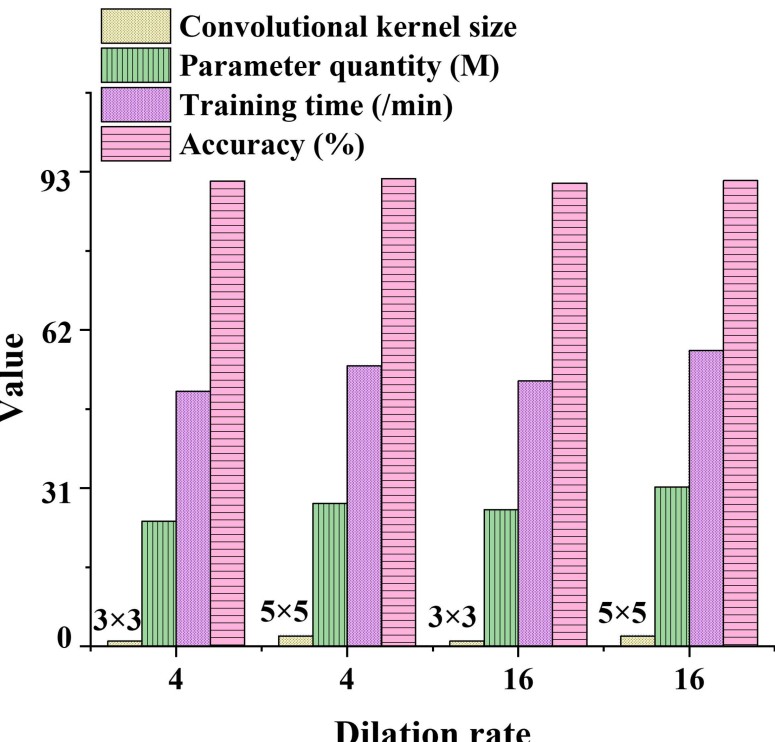

**Fig 10. The Influence of learning rates and batch sizes on the model's classification results.**

**Table 5. The Influence of dilation rates and convolution kernel size on the model's classification results.**

| Dilation rate | Convolutional kernel size | Parameter quantity (M) | Training time (/min) | Accuracy (%) |
|---|---|---|---|---|
| 4 | 3x3 | 24.5 | 50 | 91.2 |
| 4 | 5x5 | 28 | 55 | 91.7 |
| 16 | 3x3 | 26.8 | 52 | 90.8 |
| 16 | 5x5 | 31.2 | 58 | 91.3 |

## 5. Conclusions

With the development of DL technology, research on environmental art image classification has gradually gained attention. The proposed DKSE-Net model demonstrates its efficient classification capability in complex cultural backgrounds in theory. Meanwhile, it achieves significant effects in environmental art image classification tasks, with a classification accuracy of 92.7%. This performance not only reflects the model's excellent performance on the dataset but also indicates its great potential in practical applications. Especially in artwork classification and recommendation systems, DKSE-Net can effectively handle complex cultural backgrounds, helping users better explore and understand artworks. It is believed that with continuous optimization of the model, DKSE-Net can play an important role in multiple practical scenarios. The practical applications of DKSE-Net may include automatic classification and recommendation of artworks, digital protection of cultural heritage, enhancement of art appreciation in the education field, and analysis and trend forecasting in the art market. By accurately identifying and classifying environmental art images, DKSE-Net can provide strong support for research and practice in related fields. In summary, the DKSE-Net model exhibits

outstanding performance in environmental art image classification and has significant practical application value in multiple domains.

Despite the achievements, this study still has some limitations. Firstly, the training dataset used may be limited in terms of cultural diversity, and future research could consider introducing more diverse image data to enhance the model's generalization capability. In addition, DKSE-Net may face performance degradation when dealing with extremely complex scenes, so incorporating more advanced adversarial training and optimized attention mechanisms may be a critical direction for future research. Future research could explore the following directions. First, the architecture of DKSE-Net should be optimized to improve its classification performance on different types of images; Second, more external datasets must be introduced to enrich training samples and enhance the model's robustness; Third, it is necessary to consider combining other DL techniques and algorithms to expand the model's applicability.

## Supporting information

**S1 Data. Dataset.**
(ZIP)

## Author contributions

**Conceptualization:** Chenchen Liu.

**Data curation:** Chenchen Liu, Haoyue Guo.

**Formal analysis:** Chenchen Liu.

**Investigation:** Chenchen Liu.

**Methodology:** Haoyue Guo.

**Project administration:** Chenchen Liu, Haoyue Guo.

**Resources:** Chenchen Liu, Haoyue Guo.

**Software:** Chenchen Liu.

**Supervision:** Chenchen Liu, Haoyue Guo.

**Validation:** Haoyue Guo.

**Visualization:** Chenchen Liu.

**Writing – original draft:** Chenchen Liu, Haoyue Guo.

**Writing – review & editing:** Chenchen Liu, Haoyue Guo.

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
