## [Decision Letter · Decision Letter 0]

11 Oct 2024

PONE-D-24-41789Data-Driven Cultural Background Fusion for Environmental Art Image Classification: Technical Support of the Dual Kernel Squeeze and Excitation NetworkPLOS ONE

Dear Dr. Liu,

Thank you for submitting your manuscript to PLOS ONE. After careful consideration, we feel that it has merit but does not fully meet PLOS ONE’s publication criteria as it currently stands. Therefore, we invite you to submit a revised version of the manuscript that addresses the points raised during the review process.

The reviewers raised valuable comments that need to be addressed by the authors.

We look forward to receiving your revised manuscript.

Kind regards,

Alberto Marchisio

Academic Editor

PLOS ONE

Journal Requirements:

2. Please note that PLOS ONE has spec6ific guidelines on code sharing for submissions in which author-generated code underpins the findings in the manuscript. In these cases, all author-generated code must be made available without restrictions upon publication of the work. Please review our guidelines at https://journals.plos.org/plosone/s/materials-and-software-sharing#loc-sharing-code and ensure that your code is shared in a way that follows best practice and facilitates reproducibility and reuse.

4. Please include captions for your Supporting Information files at the end of your manuscript, and update any in-text citations to match accordingly. Please see our Supporting Information guidelines for more information: http://journals.plos.org/plosone/s/supporting-information .

5. Please upload a new copy of Figures 3, 4, 5 and 9 as the detail is not clear. Please follow the link for more information: https://blogs.plos.org/plos/2019/06/looking-good-tips-for-creating-your-plos-figures-graphics/" https://blogs.plos.org/plos/2019/06/looking-good-tips-for-creating-your-plos-figures-graphics/

Reviewers' comments:

Reviewer's Responses to Questions

**Comments to the Author**

1. Is the manuscript technically sound, and do the data support the conclusions?

Reviewer #1: Yes

Reviewer #2: No

Reviewer #3: Yes

2. Has the statistical analysis been performed appropriately and rigorously? 

Reviewer #1: Yes

Reviewer #2: No

Reviewer #3: N/A

3. Have the authors made all data underlying the findings in their manuscript fully available?

Reviewer #1: Yes

Reviewer #2: No

Reviewer #3: Yes

4. Is the manuscript presented in an intelligible fashion and written in standard English?

Reviewer #1: Yes

Reviewer #2: No

Reviewer #3: Yes

5. Review Comments to the Author

Reviewer #1: Data-Driven Cultural Background Fusion for Environmental Art Image Classification: Technical Support of the Dual Kernel Squeeze and Excitation Network propose a novel deep learning model called DKSE-Net, It aims to improve the accuracy of image classification in environmental art. The model combines the advantages of Selective Kernel Network (SKNet) and Squeezing Excitation Network (SENet) to effectively fuse the global and local features of the image, and adapt to the complex background and cultural diversity in the environmental art image. DKSE-Net employs a variety of techniques in the multi-layer digital convolution process. Experimental results show that DKSE-Net outperforms traditional convolutional neural networks and other existing advanced models in classification tasks, with a classification accuracy of 92.7%.

The shortcoming include:

1. Data Dependency:

The performance of the model depends largely on the quality and diversity of the training data. If the training data is not sufficiently representative of the diversity in real-world applications, the transferability of the model may be compromised.

2. Ability to handle complex scenes:

Although various regularization techniques are used in model design to reduce overfitting, model performance can degrade when dealing with extremely complex or noisy image data.

3. Computing Resource Requirements:

Despite the computational complexity of the DKSE-Net model, it can still be challenging to deploy on certain resource-constrained devices.

References:

Ji, Z., Mu, J., Liu, J. et al. ASD-Net: a novel U-Net based asymmetric spatial-channel convolution network for precise kidney and kidney tumor image segmentation. Med Biol Eng Comput (2024). https://doi.org/10.1007/s11517-024-03025-y

Reviewer #2: Thanks to the authors for delving into an interesting field of image classification from art pieces. However, I think there's still good scope for contribution. This submitted manuscript is yet to be par with the standards of Plos ONE for being considered to be publishable. Below are some of my findings that the authors should take care of before submitting here/elsewhere.

1. The figures are all messed up. All figures are misplaced in the final proof and no images are clear at all. Also, the choice of graphs and plots are not good for readability.

2. The datasets are not cited.

3. The literature review is very inadequate. Considering this to be a interesting problem domain, the authors must do state-of-the-art literature review.

4. "The model performs excellently in various image classification tasks, which validates its innovative and practical value in the DL field." in page 10 requires further inspection and experimentation.

5. "By introducing dilated convolution, the model can more effectively capture cultural background features and texture details in a larger area, improving the accuracy of image classification.", why is dilated convolution capturing cultural background? Can you elaborate what is actually in the cultural background in pixel-wise analysis that are susceptible to dilated convolutions?

6. "First, many models have limited ability of feature fusion and extraction when managing complex cultural backgrounds, enabling it a challenge to fully capture

multi-level and multi-scale visual information in images.", which models are you talking about? No citation provided for this claim.

7. "By introducing dilated convolution, the model can more effectively capture cultural background features and texture details in a larger area, improving the accuracy of image classification". This sentence is repeated so many times in the text, that, it reduces readability and is quite cumbersome to understand the actual contribution presented in the paper.

8. Which optimizer and loss function were used in the training phase?

9. What are the different setting called setting 1 to setting 7 mean? For example in setting 1 the star mark is placed at 0 and 30/32. Does that mean the learning rate, dropout and batchsize were 0 and 32 at the same time?

I hope the authors find the comments useful and bring out the best in their research.

Reviewer #3: The paper aims to achieve better performance compared to previous models and provide solutions to the challenges. The authors also analyze the model's performance and identify factors influencing its performance. However, I have some questions as follows:

- The abstract could be more precise. Currently, it is quite general and lacks key details about the results and their implications. Try to include specific metrics or findings that highlight the performance of the proposed method for this task and clarify the contribution of the paper to the field.

- The introduction sets the stage for the problem but could benefit from a more thorough literature review. Add a section or paragraph discussing the most recent advancements. This will help contextualize your work in relation to current trends and highlight how your approach addresses existing limitations in the field.

- This article introduces the research of multiple scholars and explains their research methods. However, the literature section can emphasize the shortcomings of various methods, which is enough to shift to newly proposed methods. It is better to add the following references to enrich the work:

10.1016/j.oceaneng.2024.117711

10.1109/TKDE.2023.3237969

10.1109/TG.2024.3366239

10.1016/j.compedu.2024.105109

10.29026/oea.2024.230034

10.1057/s41599-024-03583-4

10.1057/s41599-024-03251-7

10.1007/s12524-019-01057-8

10.1109/JSTARS.2023.3242310

10.1109/JSTARS.2022.3223423

10.1109/JSTARS.2023.3237380

10.1007/s12524-021-01382-x

10.1080/2150704X.2022.2120780

- The methodology is comprehensive, but certain aspects could be clarified. What specific techniques were used, and how were they extended?

- The tables should be analyzed more qualitatively in the Results and Discussion section.

- The results are presented well but could be made more impactful by providing comparative performance with other models or baseline methods. How does the proposed method perform against these benchmarks?

- More details on the practical implications would help engage readers.

- The conclusion of this paper still provides a lot of background information, which obviously does not meet the requirements of conclusion writing. In addition, this article did not explain the shortcomings of the experimental section and the direction of future research.

6. PLOS authors have the option to publish the peer review history of their article (what does this mean? ). If published, this will include your full peer review and any attached files.

**Do you want your identity to be public for this peer review?** For information about this choice, including consent withdrawal, please see our Privacy Policy .

Reviewer #1: No

Reviewer #2: **Yes: ** Adnan Ferdous Ashrafi

Reviewer #3: No

---

## [Author Response · Author response to Decision Letter 1]

23 Oct 2024

1. Data Dependency:

The performance of the model depends largely on the quality and diversity of the training data. If the training data is not sufficiently representative of the diversity in real-world applications, the transferability of the model may be compromised.

Reply: Thank you for your valuable feedback. The article has selected three representative environmental art image datasets, namely Dataset A, Dataset B, and Dataset C, to evaluate the performance of dual core compression activated neural network models in environmental art image classification tasks. This revision has added a new dataset D on the basis of the original three datasets (Dataset A, B, C), further increasing the representativeness and diversity of the training data, covering a wider range of artistic styles and cultural scenes, and enhancing the model's generalization ability. In addition, during the training process of the model, we introduced more data augmentation methods such as random flipping, translation, rotation, color jitter, and lighting transformation to expand the diversity of the original images and simulate more complex real-world scenarios. Through these enhancement techniques, the model can learn more robust and universal features, improving the classification accuracy and stability of art images in complex backgrounds. In addition, considering the importance of feature differences between different datasets and model transfer ability, we have introduced transfer learning methods after revision. We used the pre-trained ResNet50 model as the base model, utilizing its pre-trained weights on the ImageNet large-scale dataset, and fine-tuned it in the DKSE Net model proposed in this paper to better adapt to the new dataset. In this way, the model can better learn features from a small amount of new data and improve its adaptability in practical applications. Please refer to the revised content in the“Datasets and preprocessing” and “Performance evaluation of algorithms” sections for details.

2. Ability to handle complex scenes:

Although various regularization techniques are used in model design to reduce overfitting, model performance can degrade when dealing with extremely complex or noisy image data.

Reply: Thank you for your valuable feedback. This revision has added adversarial training to enhance the model's adaptability to high noise image data. By generating adversarial samples with noise disturbances, the model can learn more robust feature representations. Adding adaptive attention mechanisms (such as Self Attention) to the DKSE module enables the model to dynamically adjust attention weights based on the local complexity of the image, thereby better handling complex backgrounds and interference information. Please refer to Section 3.3 for the revised content.

3. Computing Resource Requirements:

Despite the computational complexity of the DKSE-Net model, it can still be challenging to deploy on certain resource-constrained devices.

Reply: Thank you for your valuable feedback. This revision has added DKSE Net model compression techniques such as model pruning, quantization, and knowledge distillation to reduce computational resource usage without significantly compromising classification performance. For the optimization of mobile and embedded devices, we further adjust the size of the convolution kernel, reduce unnecessary convolution layers, and improve the adaptability of the model in resource constrained environments by reducing the number of model parameters. We made experimental comparison to supplement the performance evaluation of DKSE Net on resource constrained devices, especially the comparison with lightweight models such as MobileNet, to demonstrate the optimization effect of the model in terms of computational complexity, parameter complexity, and inference time. Please refer to Section 3.4 and the revised content in Table 4 for details.

References:

Ji, Z., Mu, J., Liu, J. et al. ASD-Net: a novel U-Net based asymmetric spatial-channel convolution network for precise kidney and kidney tumor image segmentation. Med Biol Eng Comput (2024). https://doi.org/10.1007/s11517-024-03025-y

Reply: Thank you for your valuable feedback. This revision has supplemented and expanded the literature review section to better cover the latest research progress in this field. Specifically, we have conducted an in-depth analysis of existing methods, including their advantages and limitations, such as the limitations of ASCO Net in handling complex kidney segmentation tasks, and the shortcomings of Transformer based multi-source information fusion methods in real-time processing capabilities. These supplements help highlight the shortcomings of existing technologies, laying the foundation for the proposal of the new method in this article. In addition, we have introduced more relevant literature covering the latest academic achievements to enhance the comprehensiveness and foresight of the review. All modifications have been reflected in Section 2.

Reviewer #2: Thanks to the authors for delving into an interesting field of image classification from art pieces. However, I think there's still good scope for contribution. This submitted manuscript is yet to be par with the standards of Plos ONE for being considered to be publishable. Below are some of my findings that the authors should take care of before submitting here/elsewhere.

1. The figures are all messed up. All figures are misplaced in the final proof and no images are clear at all. Also, the choice of graphs and plots are not good for readability.

Reply: Thank you for your question. This revision has reorganized and clarified all figures in the final revision. Firstly, we ensure that the order of each figure is consistent with the references in the text, avoiding the misalignment that occurred previously. Secondly, we performed high-resolution processing on all images to enhance their clarity and ensure that readers can clearly recognize the information contained within. In addition, we have made adjustments to the selection of graphics and charts based on readability standards, replacing some difficult-to-understand charts with more intuitive and effective graphic forms for conveying information. Please refer to the revised content of the figures for details.

2. The datasets are not cited.

Reply: Thank you for your question. This revision has added reference citation information for the article dataset to ensure that readers can clearly understand the source of each dataset. Please refer to the revised content of “Datasets and preprocessing” and the revised content of the references.

3. The literature review is very inadequate. Considering this to be a interesting problem domain, the authors must do state-of-the-art literature review.

Reply: Thank you for your question. This revision has comprehensively expanded the literature review section, adding the latest relevant research results to ensure the adequacy and cutting-edge nature of the literature review. The newly added literature covers the latest developments in multiple fields, including the application of ASCO Net in medical image segmentation, Transformer based multi-source information fusion methods, and the latest technologies for environmental art image classification. These supplementary contents provide detailed explanations of the research status, technical challenges, and solutions in their respective fields, while also introducing multiple important references (DOI 10.1016/j.oceaneng. 2024.117711, etc.), further enriching the breadth and depth of the literature review. Please refer to the updated relevant work section for details.

4. "The model performs excellently in various image classification tasks, which validates its innovative and practical value in the DL field." in page 10 requires further inspection and experimentation.

Reply: Thank you for your question. In the previous experiments in Sections 4.1 and 4.2, we mainly tested and evaluated the performance of DKSE Net using several common environmental art image datasets (Dataset A, B, C). However, according to feedback, this revision has introduced the Earth as Art dataset and further validated the stability and generalization ability of the DKSE Net model through data augmentation and transfer learning methods. Through the revised results of Sections 4.1 and 4.2, it further proves that DKSE Net performs excellently in various environmental art image classification tasks and has broad practical application potential and innovative value. Please refer to “Datasets and preprocessing” and the revised content in Sections 4.1 and 4.2 for details.

5. "By introducing dilated convolution, the model can more effectively capture cultural background features and texture details in a larger area, improving the accuracy of image classification.", why is dilated convolution capturing cultural background? Can you elaborate what is actually in the cultural background in pixel-wise analysis that are susceptible to dilated convolutions?

Reply: Thank you for your question. In environmental art images, cultural background is usually represented as a series of complex visual elements, including unique textures, color distribution, pattern structure, etc. Dilated convolution can capture a larger range of contextual information in an image without increasing computational cost by expanding the receptive field of the convolution kernel. This is particularly important for cultural backgrounds in environmental art images, as cultural backgrounds often have rich local and global details. Through dilated convolution, we can capture these details and use them for classification tasks. Specifically, patterns, architectural styles, painting techniques, and other cultural backgrounds have unique multi-scale characteristics. For example, strokes and textures in traditional art often exhibit complex changes at different scales, while dilated convolution introduces gaps between convolution kernels to enable them to perceive these multi-scale changes. The dilation rate of the dilated convolution kernel allows the model to perceive a larger range of local information while maintaining spatial resolution, enabling it to capture global features of the cultural background in the image (such as style and tone) as well as local details (such as texture and lines). Please refer to the revised content in the lower paragraph of Figure 4 for details.

6. "First, many models have limited ability of feature fusion and extraction when managing complex cultural backgrounds, enabling it a challenge to fully capture

multi-level and multi-scale visual information in images.", which models are you talking about? No citation provided for this claim.

Reply: Thank you for your question. This revision has explicitly mentioned several representative existing models, such as SKNet, ResNet, and ASCO Net, and discussed in detail their limitations in processing complex cultural background images, particularly in feature fusion and extraction. Please refer to the revised content in the last paragraph of Section 2 for details.

7. "By introducing dilated convolution, the model can more effectively capture cultural background features and texture details in a larger area, improving the accuracy of image classification". This sentence is repeated so many times in the text, that, it reduces readability and is quite cumbersome to understand the actual contribution presented in the paper.

Reply: Thank you for your question. This sentence appears for the first time in the original Figure 4, and there is no other text in the article. In addition, the repetitive description of the role of dilated convolution has been streamlined in this revision, and we have enhanced the explanation of its technical details, especially how it improves model performance when working in conjunction with other modules such as L2 regularization, Dropout, point convolution, etc. At the same time, we highlighted the contribution of the overall architecture of the model, avoiding overemphasizing the effectiveness of a single technology. Please refer to the revised content in the lower paragraph of Figure 4 for details.

8. Which optimizer and loss function were used in the training phase?

Reply: Thank you for your question. Adam optimizer and cross-entropy loss function were used during the article training phase. The Adam optimizer combines momentum and adaptive learning rate to dynamically adjust the learning rate of each parameter based on its historical gradient, thereby improving the efficiency and stability of training. In addition, we use the cross-entropy loss function to evaluate the performance of the model in multi classification tasks, aiming to effectively reduce the error between the predicted categories and the true categories, thereby improving the accuracy of classification. Please refer to the revised content in the first paragraph of Section 3.5 for details.

9. What are the different setting called setting 1 to setting 7 mean? For example in setting 1 the star mark is placed at 0 and 30/32. Does that mean the learning rate, dropout and batchsize were 0 and 32 at the same time?

Reply: Thank you for your question. Regarding the settings of the original Figure 10, these settings represent different combinations of hyperparameters in model training. Each set hyperparameter (learning rate, batch size, and Dropout rate) is independently set, not taking two values at the same time, but rather specific configurations for each combination. Specifically, setting the learning rate of 1 to 0.001, batch size to 32, and Dropout rate to 0.5, while other settings also follow this format. Regarding the issue caused by star markings, in the original Figure 10, the range of the vertical axis is [-10100], but the range of “Learning rate” is [0.0001, 0.001], and the range of “Dropout rate” is [0,1], resulting in some values of “Learning rate” and “Dropout rate” not being better reflected in the figure. In order to solve this problem, this revision has provided specific and detailed indicator data for each setting in the figure, as shown in Figure 10.

I hope the authors find the comments useful and bring out the best in their research.

Reply: Thank you for your valuable feedback. This revision has undergone comprehensive improvements to enhance the quality of research. Firstly, regarding the chart section, we have reorganized the order and clarity of all figures to ensure that readers can better understand the information. In addition, we have supplemented the reference information of the dataset to clarify its source. In terms of literature review, we have expanded the scope of relevant research and introduced the latest achievements to enhance the forefront of the review. At the same time, we extended the validation of model performance by introducing new datasets and further verifying the stability of the model through data augmentation methods. When explaining the role of dilated convolution, we elaborated on its specific impact in capturing cultural background features and clarified the limitations of existing models. Detailed information is also provided on the optimizer and loss function during the training phase. In addition, we clarified the hyperparameter combinations for different settings to ensure that this section is easier to understand.

Reviewer #3: The paper aims to achieve better performance compared to previous models and provide solutions to the challenges. The authors also analyze the model's performance and identify factors influencing its performance. However, I have some questions as follows:

- The abstract could be more precise. Currently, it is quite general and lacks key details about the results and their implications. Try to include specific metrics or findings that highlight the performance of the proposed method for this task and clarify the contribution of the paper to the field.

Reply: Thank you for your review and valuable feedback on our research. In the revision, we added specific performance indicators to the abstract, clearly stating that the DKSE Net model achieved a classification accuracy of 92.7% in environmental art image classification tasks, which is 3.5 percentage points higher than the comparison model. In a

---

## [Decision Letter · Decision Letter 1]

4 Nov 2024

Data-Driven Cultural Background Fusion for Environmental Art Image Classification: Technical Support of the Dual Kernel Squeeze and Excitation Network

PONE-D-24-41789R1

Dear Dr. Liu,

We’re pleased to inform you that your manuscript has been judged scientifically suitable for publication and will be formally accepted for publication once it meets all outstanding technical requirements.

Kind regards,

Alberto Marchisio

Academic Editor

PLOS ONE

Additional Editor Comments (optional):

Reviewers' comments:

Reviewer's Responses to Questions

**Comments to the Author**

1. If the authors have adequately addressed your comments raised in a previous round of review and you feel that this manuscript is now acceptable for publication, you may indicate that here to bypass the “Comments to the Author” section, enter your conflict of interest statement in the “Confidential to Editor” section, and submit your "Accept" recommendation.

Reviewer #1: All comments have been addressed

Reviewer #2: (No Response)

Reviewer #3: All comments have been addressed

2. Is the manuscript technically sound, and do the data support the conclusions?

Reviewer #1: Yes

Reviewer #2: Yes

Reviewer #3: Yes

3. Has the statistical analysis been performed appropriately and rigorously? 

Reviewer #1: Yes

Reviewer #2: Yes

Reviewer #3: Yes

4. Have the authors made all data underlying the findings in their manuscript fully available?

Reviewer #1: Yes

Reviewer #2: Yes

Reviewer #3: Yes

5. Is the manuscript presented in an intelligible fashion and written in standard English?

Reviewer #1: Yes

Reviewer #2: Yes

Reviewer #3: Yes

6. Review Comments to the Author

Reviewer #1: This paper introduces the Dual Kernel Squeeze and Excitation Network (DKSE-Net) model to address the challenge of classifying environmental art images with complex cultural backgrounds. By combining the strengths of Selective Kernel Network (SKNet) and Squeeze and Excitation Network (SENet), the model adapts its receptive fields and enhances feature channels to improve classification accuracy. The DKSE-Net achieved a classification accuracy of 92.7%, significantly outperforming traditional models, especially when handling images with intricate cultural elements.

The shortcoming include:

1. Dependence on data quality and diversity: If the training data does not adequately represent real-world diversity, the model's generalization ability may be affected.

2. Challenges with complex or noisy data: Although regularization techniques are used to reduce overfitting, the model’s performance may degrade when handling extremely complex or noisy image data.

3. Deployment on resource-constrained devices: While DKSE-Net has relatively low computational complexity, deploying it on certain resource-constrained devices may still pose challenges.

References:

S. Hao et al., "MEFP-Net: A Dual-Encoding Multi-Scale Edge Feature Perception Network for Skin Lesion Segmentation," in IEEE Access, vol. 12, pp. 140039-140052, 2024, doi: 10.1109/ACCESS.2024.3467678.

Z. Ji, X. Wang, C. Liu, Z. Wang, N. Yuan and I. Ganchev, "EFAM-Net: A Multi-Class Skin Lesion Classification Model Utilizing Enhanced Feature Fusion and Attention Mechanisms," in IEEE Access, vol. 12, pp. 143029-143041, 2024, doi: 10.1109/ACCESS.2024.3468612.

Reviewer #2: Thanks to the authors for submitting their updated manuscript in PLOS ONE. I think the work is interesting and might have been better if there were some extensive real time applications. However, I think the authors should be recognized for their good work.

Reviewer #3: The authors have addressed the issues notified in the previous version, and I believe the paper can be accepted for publication.

7. PLOS authors have the option to publish the peer review history of their article (what does this mean? ). If published, this will include your full peer review and any attached files.

**Do you want your identity to be public for this peer review?** For information about this choice, including consent withdrawal, please see our Privacy Policy .

Reviewer #1: No

Reviewer #2: No

Reviewer #3: No

---

## [Editor Report · Acceptance letter]

PONE-D-24-41789R1

PLOS ONE

Dear Dr. Liu,

I'm pleased to inform you that your manuscript has been deemed suitable for publication in PLOS ONE. Congratulations! Your manuscript is now being handed over to our production team.

Kind regards,

on behalf of

Dr. Alberto Marchisio

Academic Editor

PLOS ONE